# A Simple Validated Method for the Estimation of Pepsin Activity in Microtiter Array for the INFOGEST Protocol

**DOI:** 10.3390/foods12203851

**Published:** 2023-10-20

**Authors:** Maximiliano Ramm, Bárbara Alarcón-Zapata, Juan Monsalves, Luis Bustamante

**Affiliations:** Departamento de Análisis Instrumental, Facultad de Farmacia, Universidad de Concepción, Concepción 4030000, Chile

**Keywords:** INFOGEST, pepsin, Folin–Ciocalteu reagent, microtiter array, assay miniaturization, enzyme activity

## Abstract

The INFOGEST protocol has been widely used as a static *in-vitro* simulation of gastrointestinal food digestion for bioaccessibility assessments on bioactive compounds. The standardization of the activity of several enzymes, such as pepsin, via UV-spectrophotometry of digested hemoglobin at 280 nm is a key step in the protocol. Standardization is a crucial stage since it is necessary to determine the quantity of enzyme to be added to the sample for digestion. However, this method is yet to be analytically validated; it requires quartz cuvettes and large volumes of samples and is time-consuming. Thus, we reviewed and adapted a well-known colorimetric method in microplates array by using the Folin–Ciocalteu reagent, and this study is the first to report for miniaturization of this method, the advantages of which include its automation, ease of use, the low volume of samples required, the minimal use of reagents, and speed. This method was compared to the traditional UV method, and the comparison results show no statistical difference between the inter day means for each group (*p* > 0.05). The proposed method was validated, showing high reproducibility (8% as inter-day CV) and statistically comparable results with the traditional UV spectrophotometric method.

## 1. Introduction

The INFOGEST protocol is a harmonized static method that simulates the physiological conditions of the upper gastrointestinal tract to test the digestion of foods and pharmaceuticals. It was developed by COST INFOGEST, an international multidisciplinary network emphasizing the sharing of knowledge on the digestive process by identifying the beneficial food components released in the gut during digestion and their beneficial effect on human health [1].

The protocol consists of several phases: the preparation of the solutions, the oral stage, the gastric stage, the intestinal stage, and subsequent sample analysis. In the preparation phase, the fluid stock solutions (oral, gastric, and intestinal) and the enzyme solutions (amylase, lipase, pepsin, and pancreatin) are prepared. A critical step to be considered is the estimation of enzyme activities, which is carried out mainly via UV spectrophotometric methods (as described by Minekus [2]). Pepsin activity is estimated by increasing enzyme concentrations to a standardized concentration of acidified hemoglobin. Digestion is performed under controlled temperature, agitation, and time conditions. Then, the centrifuged supernatant is collected and analyzed via absorptiometry at 280 nm.

Pepsin activity estimation was initially developed as a colorimetric method using the Folin–Ciocalteu (FC) reagent [3]. Later, this method was adapted to estimate the activity of other proteases, like trypsin, papain, and cathepsin, with hemoglobin as a substrate and the FC reagent [4]. Later, other methods were studied, including the use of a p-nitrophenyl sulfite substrate (ideal for gastric secretions) [5] and the last INFOGEST-adopted UV protocol, which is performed at 280 nm directly (without any chromogenic reagent) [2].

There are crucial steps that need to be followed to correctly estimate enzymatic activity, such as the accurate weighing of the enzyme and its substrate, adequate setting of pH and temperature conditions for reactions, and its proper completion [6]. Moreover, the substrate (or product) determination for the activity estimation, being a quantitative analysis, should be a validated method to ensure the reliability of the results, considering, i.e., linearity, reproducibility, quantification limit, and sensitivity [7]. Loss of control or the incorrect calculation of these parameters could result in inaccurate activity estimation and the loss of costly reagents such as enzymes, leading to possible experimental variations in the subsequent INFOGEST digestion technique [8]. Thus, it is ideal to validate, minimize, and optimize the volume of reagents and analysis time. One of the strategies one could adopt could be miniaturization through the employment of microtiter readers for the analysis of the supernatant. These methods have been successfully applied to the estimation of phenolic content in foods [9,10], reducing sugars [11], chloride ions [12], etc. They can simultaneously handle numerous samples with only a small volume of reagents and samples in a relatively short time. There are no scientific publications on the miniaturization of this method.

Using a colorimetric test in the visible range and the FC reagent, which reacts with molecules containing phenol groups, such as amino acids and peptide residues, we compared and validated the actual UV method and proposed a miniaturized approach for the estimation of pepsin activity. This method is intended for application to pepsin activity estimation in the INFOGEST protocol but could eventually be used to estimate the activity of other proteases.

## 2. Materials and Methods

### 2.1. Materials

For the digestion process, 2 mL microcentrifuge tubes and a Thermomixer C (Eppendorf, Hauppage, NY, USA) were used. The samples were centrifuged using a Heraeus fresco 17 centrifuge (Thermo Fischer Scientific, Waltham, MA, USA) at 19 °C. For the ultraviolet spectrophotometric method (UV method), a UV-Mini 1240 (Shimadzu, Kyoto, Japan) was used with 1 cm quartz cuvettes, while for the FC spectrophotometric method (VIS method), a Synergy HTX multi-mode reader (BioTek Instruments Inc., Whiting, VT, USA) with 96-well clear polystyrene microplates (Trueline, Gurugram, India) was used. pH adjustement was performed using a pH meter HANNA HI 9017 equipped with a HI 1330 combined electrode (Hanna Instruments, Woonsocket, RI, USA).

### 2.2. Reagents

Hemoglobin from bovine blood, trichloroacetic acid (TCA) (>99%), L-tyrosine (L-Tyr) (>98%, HPLC grade), and pepsin from porcine gastric mucosa (EC 3.4.23.1, ≥2500 unit/mg protein) were purchased from Sigma-Aldrich (Darmstadt, Germany). FC phenol reagent, sodium bicarbonate, sodium chloride, tris hydrochloride (Tris-HCl), hydrochloric acid (HCl), and sodium hydroxide (NaOH) pellets were obtained from Merck (Darmstadt, Germany). Ultra-pure water was used (Millipore Simplicity UV, Merck). For the preparation of the reagents, see Appendix A.

### 2.3. L-Tyrosine Calibration Curve

L-Tyr solutions were prepared in 10 mM HCl at concentrations ranging from 0.11 to 1.10 mM for the UV method and at concentrations ranging from 0.014 to 0.31 mM for the VIS method. A total of 10 mM HCl was used as blank. For more details, please go to Appendix A.

### 2.4. Pepsin Activity Assay

The assay was performed as detailed in the INFOGEST Appendix A with minor modifications [2]. Briefly, 500 µg/mL pepsin stock solution was prepared in 150 mM NaCl and 10 mM Tris-HCl. Six concentrations of pepsin (5, 10, 15, 20, 25, and 30 µg/mL) were prepared in 1 mL of 10 mM HCl and kept on ice. Additionally, 2% *w*/v hemoglobin stock solution was prepared by dispensing 500 µg of hemoglobin in 20 mL ultra-pure water followed by acidification with 2.5 mL of 300 mM HCl, adjusting the pH at 2.00 (±0.01) with 1 M NaOH, and completed with water to a final volume of 25 mL in a volumetric flask. 

Twelve hemoglobin solutions (6 for test tubes and 6 for blank tubes) were prepared by adding 500 µL of the stock solution to a microcentrifuge tube and incubating it for 3 min at 37 °C. A total of 100 µL of the corresponding pepsin solution was added to the test tubes, and then the whole solution was incubated at 37 °C for 10 min at 650 RPM. Rapidly, 1 mL of 5% TCA solution was added to each of the twelve solutions, and 100 µL of the pepsin concentration was added to the matching blank tubes. The resultant solutions were centrifuged at 6000× *g* for 30 min, and then the supernatant was collected for posterior analyses. Check Appendix A for details on the digestion procedure.

### 2.5. UV Spectrophotometric Method

The supernatant was analyzed directly using a UV-Spectrophotometer at 280 nm with 1 cm quartz cuvettes.

### 2.6. Proposed Miniaturized VIS Method

A 96-well microtiter plate was used. A total of 50 µL of the collected supernatant was added to each well, followed by 50 µL of 20% FC reagent and 100 µL of 6% *w*/v sodium carbonate. The absorbance of each well was measured at 760 nm after 10 min of incubation in darkness at 37 °C. For quantification in the microtiter array using the VIS method, see Appendix A; see Section IV for details on activity estimation.

### 2.7. Statistical Analysis

Simple linear regression, an F-test, and a *t*-test were carried out using GraphPad Prism V8 (GraphPad Software, San Diego, CA, USA). The differences between the absorbance of the digested and the blank at different enzyme concentration levels were plotted, and only those that met linearity criteria were considered for the average calculation. These results were compared by a two-tailed *t*-test (α = 0.95).

Figures of merit, such as linearity, the limit of detection (LOD), the limit of quantification (LOQ), and sensitivity, for the L-Tyr calibration curves were calculated using the simple linear regression slope and residual standard deviation. The LOD and LOQ were calculated using the five lowest concentrations of each curve [13].

## 3. Results

### 3.1. L-Tyrosine Calibration Curves

The hemoglobin product that is digested by the proteases is a complex mixture of TCA-soluble products. International Units (IU) are defined as the amount of enzyme that catalyzes the conversion of one micromole of substrate per minute under the specified conditions of the assay method [14]. It is impossible to track pepsin’s protease activity using hemoglobin as a substrate; nevertheless, one option is to follow the TCA-soluble products (low-molecular-weight peptides) [6]. This product has no standard solution, so L-Tyr can be considered a representative amino acid that could explain a significant part of its absorption at 280 nm and the reactivity with the FC reagent. A calibration curve was prepared for each method (UV and VIS). Table 1 summarizes the figures of merit for both methods. LOD and LOQ were determined by using the standard deviation obtained from the linear equations, only considering the five lowest points of each method calibration curve to be more representative of the smaller values’ proximities [13]. The linearity range was evaluated considering the absorbance of blanks and digested hemoglobin solutions. The VIS method demonstrates enhanced absorptivity and, consequently, greater sensitivity.

### 3.2. Pepsin Activity Estimation by INFOGEST Protocol

The estimated activity of the enzyme was calculated via the INFOGEST protocol using the following equation:(1)$$units/mg=\frac{\left(A_{Test}-A_{Blank}\right)\times 1000}{\Delta t\times X\times 0.001}(I)$$
*A*: Absorbance of the test and blank solutions at a specific wavelength.1000: Factor to convert µg to mg of pepsin powder.0.001: Absorbance value attributed to one unit of enzymatic activity.Δ*t*: Time of reaction (generally 10 min).*X*: Amount of pepsin in the final reaction mixture in the cuvette (mg), assuming 1 mL of pepsin solution added.

As is shown in equation I, the method does not consider the activity estimation through interpolation to a calibration curve; it is only expressed in terms of the *optical density* of the supernatant solutions at a specific wavelength [15], dissimilarly to the International Unit consensus. The enzymatic activity is estimated as an average activity at different pepsin concentrations, considering a linear relationship between ∆A (A_Test_ − A_Blank_) and enzyme concentration to exclude experimental variations and ensure that one is not working above the maximum concentration.

An inconsistency can be found when interpreting the activity as units/mg with its posterior calculation since a volume variation of reagents and final volume (keeping the proportions) does not change the concentration of the product nor the *optical density*, but it does change the amount of enzyme added, leading to a significant error [2]. Therefore, an alternative option could involve expressing the activity in terms of concentration (i.e., units mL/mg) or just conducting the same assay at greater volumes, keeping the proportions of solutions, will give the same difference in absorbances; thus, it must be established that 1 mL of pepsin solution is added to the assay solutions (in concordance with previous publications in the literature) [3].

In total, 30 activity values were calculated for each method on different days. The values that were to be included in the average activity estimation met the linearity criteria each day (Table 2). For the UV method, the activity estimation results were 1929 ± 57 units/mg, and the absorbance values ranged from 0.247 to 0.939, which are in the linear range of the L-Tyr calibration curve (0.050–1.300 AU). This result is in accordance with the INFOGEST pepsin activity assay; however, it does not offer insight into its linearity range or validation criteria.

In contrast, for the VIS method, the activity units were 2046 ± 159 units/mg (Table 2). The absorbance values ranged from 0.180 to 0.354 within the linear range of the L-Tyr calibration curve (0.064–0.442 AU). For activity estimation, a factor of 4 must be multiplied to correct the concentration of enzymes in the cuvette, which is four times lower due to its dilution.

An F-test of the total data also revealed no significant difference between the group variances (*p* > 0.05). Thus, the coefficient of variance calculated for the VIS method was 12% and 10% for the UV method. A comparison of the methods was performed via the application of an unpaired two-tailed *t*-test (α = 0.95) between the average activity calculated for each day (shown in Table 2), and this ultimately resulted in no significant differences (*p* > 0.05).

### 3.3. Pepsin Activity Estimation with L-Tyrosine Equivalent

The peptide bonds in hemoglobin are the substrate of pepsin. An alternative to measuring enzymatic activity is to measure the peptide residue products of the enzymatic digestion, which are correlated with L-Tyr as an equivalent [6]. Thus, for the standardization of this reaction product, L-Tyr is used.

At 280 nm, peptide and protein absorption are explained by aromatic amino acids like tyrosine, tryptophan, phenylalanine, amide and disulfide bonds, and the heme group. In contrast, at 760 nm, the color produced using the FC reagent is attributable to the reduction of phosphomolybdic-tungstic mixed acid by primarly phenols (like tyrosine) [16], as well as other antioxidant substances [17]. Pepsin preferably cleavages proteins in bulky hydrophobic amino acid residues like tyrosine, phenylalanine, and tryptophan [18], leading to soluble peptides that can be quantified using the FC reagent [19]. A definition of pepsin activity is proposed with this criterion as the L-Tyr amount in µmol (as an equivalent digestion product) generated in 1 min per mg of the enzyme and not in terms of optical density (as previously described, one unit will produce a ΔA280 of 0.001 per minute). It has the advantage of being a traceable estimation to a standard with a calibration curve, ensuring its analytical performance. After using this estimation method, the activity values were 25.6 ± 1.98 IU/mg. However, this estimation method dramatically differs from the one used in the INFOGEST protocol; thus, a transformation is proposed. The relation between the estimation of International Units (IU) and the INFOGEST pepsin activity units is 0.0125, which was obtained as a quotient between the activity estimated by the INFOGEST method and the activity in terms of IU for the FC reagent. The following equation [2] was established to estimate the activity by the L-Tyr equivalent product obtained via the VIS method.
(2)$$units/mg=\frac{{\left[L\text{-}Tyr\right]}_{Test}-{\left[L\text{-}Tyr\right]}_{blank}}{\Delta t\times X\times 0.0125}$$
[*L-Tyr*]: L-Tyrosine concentration in the test and the blank supernatant solutions (mM) determined by the calibration curve.Δ*t*: Time of reaction (generally 10 min).*X*: Concentration of pepsin powder in the final reaction mixture (in mg/mL).0.0125: Transformation factor for converting the activity units from IU/mg to units/mg.

As expected, estimation with this formula shows equal activity values to the previous method, resulting in a value of 2046 ± 159 units/mg (CV 8%); however, it uses L-Tyr concentration under validated conditions instead of optical density.

The estimation of pepsin activity using the FC reagent was described long ago by Anson and Mirsky in 1932 [3] but with greater volumes and without analytical validation nor the actual microtiter reader technologies. Our proposed estimation method has several advantages, including the fact that there is no confusion with the concentration unit of pepsin. It has been validated using a calibration curve and figures of merit shown previously. As seen in the methodology of the proposed estimation method, the supernatant volumes needed are reduced from 1 mL to 50 µL, which can eventually be adjusted to the previous hemoglobin digestion and even reduced to the volumes of the prepared enzyme. Also, the speed and automation of the assay are improved by applying a microtiter reader, which, with a calibration curve, leads to greater reproducibility. The explicit calculus with respect to pepsin estimation can be found in the Appendix A.

## 4. Conclusions

By adapting the well-known FC reaction to the analysis of digested hemoglobin in the pepsin activity estimation in the INFOGEST protocol, a miniaturized alternative method has been established and proposed in this study; the method has been validated and has no statistically significant differences with the UV method at 280 nm. This method has the advantages of being easy to implement and fairly inexpensive, and the method can also involve the use of accessible reagents, meaning that researchers in this field can easily implement it.

The validation criteria for the UV method were not explicit in the literature for the pepsin activity estimation, and these validation criteria ought to be a requirement to include in other enzyme assays to ensure the quality of further research.

Finally, we consider it essential to review this method, as it has been used for almost 100 years to estimate pepsin activity with no actual information besides the INFOGEST protocol.

## Figures and Tables

**Table 1 foods-12-03851-t001:** Figures of merit for the L-Tyr calibration curves of the UV and VIS method.

	UV Method	VIS Method
Wavelength (nm)	280	760
Concentration levels in triplicate	8	8
R^2^	0.9998	0.9984
LOD (mM)	0.01	0.001
LOQ (mM)	0.03	0.003
Linearity range (mM)	0.03–1.10	0.003–0.078
Absorptivity (L/mmol cm)	1.18	5.12

**Table 2 foods-12-03851-t002:** Average pepsin activity (units/mg) for each day in the UV and VIS methods using the INFOGEST protocol equation. ^a^: CV inter-day coefficient of variation.

	UV Method	VIS Method
n	28	25
Day 1	2013	2298
Day 2	1925	1980
Day 3	1925	2128
Day 4	1939	1923
Day 5	1844	1901
Average	1929	2046
CV % ^a^	3	8

## Data Availability

The detailed data presented in this study are available in Appendix A.

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
