# Peer review of "A Simple Validated Method for the Estimation of Pepsin Activity in Microtiter Array for the INFOGEST Protocol"

_foods, 2023, doi:10.3390/foods12203851_

Round 1

Reviewer 1 Report

This manuscript titled “A simple validated method for the estimation of pepsin activity in microtiter array for the INFOGEST protocol” report on a colorimetric method in microplates array for method miniaturization and optimization. The study is meaningful for determining the enzyme quantity to be added to the sample for digestion, but there are some drawbacks in the manuscript. So, authors should make a major revision to their manuscript.

Detailed comments

1\The specific operation process and key points of the new method should be supplemented with pictures in order to make it more intuitive and clearer.

2\The new method written in this manuscript should provide a detailed description of the specific differences in operation between the new method and traditional techniques, which can be achieved through drawing or listing in table.

3\Line 53-54, References are necessary.

4\Line 80-83, the preparation of L-Tyr solutions should be cleared, or references related to this step will be cited.

5\The use of punctuation in numbers is quite confusing, resulting in unclear sentence meanings, which should be checked and corrected.

6\Line102, the manufacturer and model of the UV spectrophotometer need to be specified.

7\Table1, the square of the R value should be written as a superscript.

8\Line149, punctuation error in document number.

9\Line159, “units' mL mg-1” Inconsistent with contextual writing.

Reviewer 2 Report

I had already read and evaluated the work. The manuscript is well prepared, clear in the exposition of the contents both as regards the methods and as regards the validation of the results.

I can only suggest minor revisions:

- specify in the objective in which fields this new methodology can be applied (lines 61-64).

- specify what is meant by "purified water" (line 89). Bidistilled or ultrapure water should be used in these cases.

-replace "RPM" with "g" or "RCF" (line 95 and 98).

Reviewer 3 Report

Comments to the authors:

The work described in the present manuscript is consistent with the scope of the journal.

Authors described a new proposal for an alternative INFOGEST protocol by using Folin Ciocalteu reagent considering a miniaturization of this method. The new approach was correctly validated and compared with the well-established method. The new methodology seems to be a valid alternative for the desired purpose.

The work is very complete but, in my opinion, some minor corrections are needed to be performed, specifically:

Major comments:

  1. Abstract, line 19: Please replace “absorptiometric” by “absorption” or “spectrophotometry”.
  2. Line 71: please specify the type of microplates used (number of wells, material, and company reference).
  3. In the excel file (supplementary material), the R2 value is highlighted in red in the sheet “pepsin activity” and this value is does not match with that of Table 1. Please clarify this point.

Minor comments:

  1. Line 62: replace “phenol” by “phenol groups”.
  2. Lines 81-82, 89-90, 165-166, 173-174, 197-198, equations (1) and (2): replace commas by dots (decimal numeration). The same in all PDF file of the supplementary material. Please check along all main manuscript.
  3. Line 93: replace “tempering” by “incubating”.
  4. Line 183-184: the sentence is confusing, please rewrite.
  5. In the supplementary material, in the figure of the last page, please replace “Absorciometry” by “absorption” or “spectrophotometry”.

Minor editing of English language required; commas must be replaced by dots.

Reviewer 4 Report

The article is devoted to the development of a new protocol for assessing peptin activity. The authors propose an adaptation of the well-known colorimetric method into a miniaturized version. The advantages of the developed method are automation, simplicity, small quantities of reagents and rapidity. The research was carried out at a high scientific level and well presented. The supplementary file provides all the necessary information to perform the technique. Thus, the article is undoubtedly of interest and can be published. However, I had some minor comments while reading the article.

1. My main question concerns significant digits, since the they determine the accuracy of the measurement. For example, line 90, "the pH adjusted at 2". This can be written if the pH was determined with litmus paper. The article does not contain information on how exactly pH was determined, but if pH was determined using a pH meter, then the accuracy of the determination should be 2.0 or 2.00. The same applies to the volume of the flask (line 91). If it is a volumetric flask, then the volume should be 25.0 ml or 25.00 depending on the accuracy of the calibration. Please check all the significant figures in section 2.4: 0.5 mg/ml or maybe 500 µg/ml? 3 minutes or 180 seconds? 10 minutes or 600 seconds?

2. In Section 2.3 and Table 1, tyrosine concentrations are expressed in micromole/mL. This is somewhat unusual for analytical chemistry. Since this is equivalent to nanomole/L, maybe it makes sense to use the generally accepted concentration unit nM? 

3. Please indicate the equipment manufacturer everywhere.

4. Please, check decimal points and commas. The article uses both simultaneously.
